# Improved economic and clinical outcomes with oritavancin versus a comparator group for treatment of acute bacterial skin and skin structure infections in a community hospital

Kimberly Saddler[1]*, Jason Zhang[2], Jennifer Sul[2], Pruthvi Patel[3], Miriams Castro-Lainez[4], Mark L. Stevens[5], Sheryl Kosler[6], Emily Lowery[7], Miguel Sierra-Hoffman[5]

1 Department of Pharmacy, DeTar Healthcare System, Victoria, Texas, United States of America, 2 Feik School of Pharmacy, University of the Incarnate Word, San Antonio, Texas, United States of America, 3 DeTar Family Medicine Residency Program, DeTar Healthcare System, Victoria, Texas, United States of America, 4 Facultad de Ciencias Medicas, Hospital Escuela Universitario, Universidad Nacional Autonoma de Honduras, Boulevard Suyapa, Tegucigalpa, Honduras, United States of America, 5 Texas A&M College of Medicine affiliated DeTar Family Medicine Residency, Victoria, Texas, United States of America, 6 DeTar Healthcare System, Victoria, Texas, United States of America, 7 University of Texas at Austin, Austin, Texas, United States of America

* kim.saddler@detar.com

**Data Availability Statement:** Data cannot be shared publicly because of associated PHI. Data

## Abstract

### Background

Oritavancin is a lipoglycopeptide antibiotic with in vitro bactericidal activity against gram-positive pathogens indicated for use in adults with acute bacterial skin and skin structure infections (ABSSSI). Its concentration-dependent activity and prolonged half-life provide a convenient single-dose alternative to multi-dose daily therapies for ABSSSI. This retrospective cohort study was conducted to quantify the clinical and economic advantages of using oritavancin compared to other antibiotic agents that have been historically effective for ABSSSI.

### Methods

Seventy-nine patients received oritavancin who had failed previous outpatient antibiotic therapy (OPAT) for cellulitis or abscess and were subsequently readmitted to the hospital as an inpatient between 2016 and 2018. These patients were compared to a cohort of 28 patients receiving other antibiotics following OPAT failure and subsequent hospitalization for these two infection types. The primary clinical end point was average length of stay (aLOS) and secondary endpoints included readmission rates for the same indication at 30 and 90 days after discharge and the average hospital cost (aHC).

### Results

A total of 107 patients were hospitalized for treatment of cellulitis or abscess. Demographic characteristics of both the oritavancin and comparator groups were similar except for the presence of diabetes. The primary clinical endpoint showed a non-significant decrease in aLOS between the oritavancin group versus comparator (2.12 days versus 2.59 days; p =

are available from the DeTar Ethics Committee (contact via Kim Saddler) for researchers who meet the criteria for access to confidential data. An alternate non-author contact for data access would be Petrina Lowery, Facility Compliance Officer (petrina.lowery@detar.com).

**Funding:** DeTar Healthcare System provided support in the form of a salary for author KS but did not have any additional role in the study design, data collection and analysis, decision to publish, or preparation of manuscript. This does not alter our adherence to PLOS ONE policies on sharing data and materials.

**Competing interests:** DeTar Healthcare System provided support in the form of a salary for author KS. Miguel Sierra-Hoffman M.D. is a member of the speaker bureau for Melinta Therapeutics. This does not alter our adherence to PLOS ONE policies on sharing data and materials.

0.097). The secondary endpoints revealed lower readmission rates associated with oritavancin treatment at 30 and 90 days; the average hospital cost was 5.9% lower for patients that received oritavancin.

## Conclusion

The results of this study demonstrate that oritavancin provides not only a single-dose alternative to multi-day therapies for skin and skin structure infections, but also a clinical and economic advantage compared to other antibiotic agents.

## Introduction

The therapeutic management of acute bacterial infections of the skin and skin structures (ABSSSIs) has been identified as a clinical area of infectious disease focus with vast opportunity for improvement for both hospital policy makers and front-line clinicians [1,2]. New pharmacological treatments are needed, but of major importance is improving cost effectiveness and efficient use of resources. In the era of value-based care, there are strong incentives to develop patient-centric treatment approaches that improve quality of care and increase efficiency while utilizing resources judiciously [2,3].

The challenge of managing ABSSSIs is complicated by its common diagnosis and healthcare burden. Miller et al examined ambulatory and inpatient data from the HealthCore Integrated Research Database between 2005 and 2010 and found that the incidence of ABSSSIs is substantial; it is approximately double the prevalence of UTI and tenfold that of pneumonia. Persons aged 45 to 64 years had the highest incidence of both ambulatory-treated and inpatient-treated ABSSSIs [4]. Tun et al performed a cross-sectional analysis of nationally representative data from the Medical Expenditure Panel Survey and found a total of 2.4 million patients experienced an ABSSSI in 2000 compared to 3.3 million in 2012, representing a 40% increase. From 2000 to 2012, the incidence of patients with at least one hospital visit for ABSSSIs increased 22%, ambulatory care visits increased 30%, and emergency department visits increased 40%. In 2012, ambulatory based ABSSSI visits accounted for the majority of all ABSSSI visits (76%), or 2.5 million visits annually–a 30% increase since 2000, followed by increases in ED visits (17%) and inpatient visits (7%) [5].

While there is sparse data on the incidence after 2012 in the U.S., a recent study by Morgan et al showed a decline of 8.0% in emergency room encounters for ABSSSIs between 2009 and 2014 [6]. Recently, Fritz et al used a nationally representative dataset from 2000–2015 and demonstrated a rise in the incidence of outpatient visits for purulent skin infections in adults, peaking in 2010–2013, followed by a plateau or slight decline [7]. Temporal trends in MRSA-related hospital-onset and community-onset infections (between 2012 and 2017) decreased by 20.5% [8]. According to our local antibiograms, produced yearly, MRSA infections have not decreased in the last three years, but continue to represent approximately 50% of Staphylococcus aureus infections. Due to continued rise in healthcare costs across the board, it is not surprising that although the rate of infection may not be increasing, the cost to treat continues to rise. These combined trends likely result in some stabilization or slight decrease in ABSSSI incidence and MRSA-related ABSSSIs.

As the incidence of ABSSSIs has remained stable over the past decade, costs have increased. Kaye et al explored data from the US Healthcare Cost and Utilization Project National Inpatient Sample between 2005 and 2011. They found that average costs and length of stay in 2011

for an adult ABSSSI inpatient were $9,895.31 and 5.0 days, respectively, primarily composed of abscess and cellulitis in 73% of patients [2]. Tun et al added further information on the impact of ABSSSI on the US healthcare system. The total estimated direct healthcare costs of ABSSSIs increased 3-fold from $4.8 billion in 2000 to $15.0 billion in 2012. Direct healthcare costs of ED visits doubled ($200 million in 2000 to $400 million in 2012) and that of inpatient visits increased 1.6-fold ($3.5 billion to $5.5 billion) [5].

This retrospective study collected data during a three-year calendar period (2016 to 2018) at DeTar Healthcare System, a 219-bed community hospital with an average daily census of 95 located in outh Texas. An oritavancin protocol was developed and approved for treatment of patients with skin and skin structure infections (see Fig 1); utilization began in August 2015. The pathway at our institution reflects a suggested practice and is not mandatory. It provides guidance on infusion requirements, pre-medication, and diluent. The pathway was not used a priori to select patients for this study. We measured the real-world experience including length of stay, readmission rate, and economic impacts of a cohort of patients who received oritavancin in an infusion center compared to a cohort of patients who received other antibiotic agents effective for cellulitis and abscess. The hypothesis of this study was that incorporation of oritavancin into hospital pathways would be associated with a reduction in hospital costs as a result of decreased average length of stay and/or decreased readmission rates for ABSSSI patients.

## Methods

This study was based on a retrospective chart review. Medical charts of 506 patients administered oritavancin in 2015 to 2018 were examined to identify patients with a clinical diagnosis of either cellulitis or abscess who were hospitalized and exhibited none of the exclusions. Patients were excluded from the study for any of the following: osteomyelitis, endocarditis, primary or secondary bacteremia, age <18 years, length of inpatient stay exceeding 7 days, any ICU stay during hospital admission, infections that required major surgical debridement and/ or wound care, and incomplete data for length of stay, readmission data to 90 days, and hospital costs. Following application of these exclusions, 79 patients remained and were the source of this analysis. These 79 patients had failed previous outpatient antibiotic therapy (OPAT) following a prior hospitalization and were treated with a single intravenous (IV) dose of oritavancin 1200 milligrams infused over 3 hours at hospital discharge and included in cohort A. These patients were compared with a cohort of 28 unique patients screened from a database of 216 patients admitted during the same period and who did not receive oritavancin (cohort B). Exclusion criteria noted above were also applied to patients in cohort B during the identification process. As with cohort A, patients in cohort B were required to have failed non-oritavancin therapy prior to hospital admission. Demographic data and clinical outcomes were obtained by chart abstraction. This study was approved by DeTar Healthcare System Ethics Committee and Chief Compliance Officer. Informed consents were waived by the Ethics Committee as Protected Health Information was not included and data was collected retrospectively. Patient data was collected via electronic medical records in May of 2019 for patients treated during the time period of August 2015 thru December of 2018. Patient data was anonymized following completion of data collection. This case study was performed in accordance with the Helsinki Declaration of 1964 and its later amendments. Waivers were approved by the hospital IRB. Hospitalization costs were obtained from medical records and hospital finance. Failure of a recent course of antibiotic therapy, either oritavancin or another course of other antibiotic(s), was defined as patients who returned to the hospital due to worsening infection symptoms or lack of improvement within 30 days or 90 days, and for oritavancin specifically within 14 days post-dose.

**DeTar HEALTHCARE SYSTEM**

Patient Sticker

**Diagnosis:** ________________________

# Orbactiv (Oritavancin) Order Form

**Indication:** Acute bacterial skin structure infections (ABSSSI) due to suspected gram-positive organisms, including MRSA.

## Criteria for treatment:

1) Infection severe enough to present to ED (Enron Class II or mild Class III)
   Class II: febrile and ill appearing but no unstable comorbidities
   Mild Class III: One or more unstable comorbidity
2) Requires IV antibiotics
3) No major comorbidities requiring 24-hour nursing care

## Exclusion Criteria:

1) Inpatients
2) Patients who can be managed sufficiently with oral therapy
3) Patients with severe hepatic impairment (Child-Pugh Class C)
4) Hypersensitivity to Oritavancin
5) Patient currently on therapeutic heparin infusion

## Restrictions:

1) Patients may not be inpatient status, they must be in outpatient status.
   (Chemo infusion clinic is not an option for infusion location)
2) Pregnancy category C
3) Breastfeeding

## Dose and Administration:

Oritavancin 1200 mg IVPB in 1000 ml of D5W over 3 hours
   (if patient develops flushing, urticaria, pruritus, slow the infusion to run over 5 hours)

Give Benadryl (diphenhydramine) 50 mg po 30 minutes prior to infusion

## Follow Up Instructions:

Follow Up Appt in 48-72 hours: _______________________________________

*Nurse Signature* ___________________*Date*_________ *Time*______ □ *Read Back & Verified*

*Physician Signature*______________________________ *Date*_________ *Time*______

12/16, 3/17,12/18

**Fig 1. Hospital protocol for administration of oritavancin.**

The evaluable safety cohort consisted of 107 patients who received oritavancin or other antibiotic therapies. Medical records were reviewed to identify drug-related treatment-emergent adverse event (TEAE) considered by the physician to be definitely related or possibly related to oritavancin or other antibiotics.

The primary endpoint was average length of stay (aLOS). The secondary endpoints included readmission rates for the same indication at 30 and 90 days after discharge and the average hospital cost (aHC). The average cost of hospitalization for patients within each cohort was derived from non-ICU medical-surgical bed charges adjusted using the hospital's cost-to-charge ratio. Cost avoidance included differences in costs of the index hospitalization plus readmissions in each cohort. The cost of readmissions was based on the greater aLOS observed in cohort B (2.59 hospital days). Cost avoidance used an average cost per hospital stay of $3,879.43 (Detar Healthcare System) and was normalized for 100 patients treated with oritavancin to compare against non-oritavancin treatments.

Data were summarized using descriptive statistics to characterize patient demographics and clinical conditions. Due to positive skewing, continuous variables for outcome measures were log-transformed to calculate average lengths of stay (aLOS) and average hospital costs (aHC) in each cohort and retransformed thereafter. Where necessary, actual costs are used for comparison. The statistical analysis plan included calculation of level of significance for aLOS comparing cohort A and cohort B cumulatively and for each year of the study. Similarly, aHC data were analyzed over the entire study period. Furthermore, the Chi-square test was used to compare the two cohorts regarding readmission status at 30 and 90 days. A significance level of 0.05 was used throughout.

## Results

A total of 107 patients were retrospectively identified and which were admitted to the inpatient medical service for failed treatment of ABSSSIs as outpatients. Treatment post-discharge was retrospectively identified as oritavancin in patients admitted for recurrent or failed ABSSSI therapy (cohort A, n = 79), or recurrent episode of ABSSSI treated with alternative agents (cohort B, n = 28), at discharge. Patient identification and cohort assignments are shown in Fig 2. All patients suffered from cellulitis or abscess.

Demographic characteristics of both the oritavancin and comparator groups were similar (Table 1) except for greater age in cohort B (median 61 years versus 51 years). Males predominated in both cohorts. The majority of patients were obese. Hypertension was observed in at least half of patients in each cohort. Cellulitis was the predominant infection with the remaining patients hospitalized for recurring or failed treatment of abscess. *Staphylococcus aureus* was the predominant pathogen in 77 percent of positive cultures; methicillin-susceptible and methicillin-resistant phenotypes were similar. Diagnosis-Related Group 603 was the discharge diagnosis for all patients in cohort B while it was the primary DRG in 53 percent of patients in cohort A.

Primary and secondary endpoints showed improvement in the oritavancin group (Table 2). The primary clinical endpoint showed a non-significant decrease in aLOS with the oritavancin group versus comparator (2.12 days versus 2.59 days; p = 0.097). Actual average ($3,959.05 versus $4,256.28, respectively; P = 0.55) and log-transformed average hospital costs ($3,376.57 versus $3,588.68; P = 0.63) were lower in the oritavancin cohort compared to cohort B, but did not reach statistical significance. The average hospital cost was lower for patients that received oritavancin (7.0% for average costs and 5.9% for log-transformed average costs). Another secondary endpoint was significant for lower readmission rates associated with oritavancin treatment within 30 and 90 days: 10.1% (8/79) and 12.7% (10/79), respectively; for the comparator

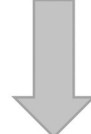

| Hospitalized patients who failed outpatient therapy for cellulitis or abscess and treated with oritavancinfollowing discharge between August 1 2015 and December 30 2018 |
|---|

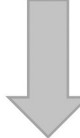

| Hospitalized patients who failed outpatient therapy for cellulitis or abscess and treated with a non-oritavancin antibiotic following discharge between August 1 2015 and December 30 2018 |
|---|

| Excluded from analysis [a] (n = 211) |
|---|

| Excluded from analysis [a] (n = 188) |
|---|

| Oritavancin single-dose therapy following discharge of patients who failed prior outpatient antibiotic therapy Cohort A (n = 79) |
|---|

| Non-oritavancin therapy following discharge of patients who failed prior outpatient antibiotic therapy Cohort B (n = 28) |
|---|

| Primary outcome of interest: Length of stay (LOS). Secondary outcomes of interest: a) readmission rates at 30-day and 90-day from end of therapy, and b) cost of hospitalization (US dollars) |
|---|

[a] Exclusion criteria: osteomyelitis, endocarditis, primary or secondary bacteremia, age <18 years, length of inpatient stay exceeding 7 days, any ICU stay during hospital admission, infections that required major surgical debridement and/or wound care, and incomplete data for length of stay, readmission data to 90 days, and hospital costs.

**Fig 2. Flow diagram of patient allocation into cohorts based on treatment regimen.**

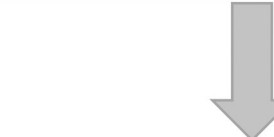

group the readmission rate was 60.7% (17/28) at both timepoints. The difference in readmission rates between the oritavancin and comparator cohorts was statistically significant for 30 days and 90 days (p<0.001 and p<0.001, respectively).

Only 6 of 79 (7.6%) patients required reevaluation within 14 days following a single intravenous (IV) dose of oritavancin 1200 milligrams delivered at discharge. Characteristics and medical narratives are described in Table 3. Two patients received a second dose of oritavancin 1200 milligrams as outpatients with subsequent resolution of infection. Two additional patients were readmitted to an inpatient medical service for unsatisfactory clinical response; one patient was discharged on non-oritavancin antibiotics but eventually required toe amputation and the second patient finally required several days of meropenem administered at a nearby long-term acute care facility. The final two patients were evaluated as outpatients

**Table 1. Patient demographics.**

| Characteristic | Oritavancin (n = 79) | Comparator (n = 28) |
|---|---|---|
| Age | | |
| Mean, yr (SD) | 51.3 (17.2) | 57.6 (21.3) |
| Median, yr | 51 | 61 |
| Range, yr | 21–93 | 25–92 |
| Age ≥ 65, no. (%) | 19 (24.1) | 10 (35.7) |
| Sex, no. (%) | | |
| Male | 49 (62) | 17 (61) |
| Race, no. (%) | | |
| Caucasian | 46 (58.2) | 15 (53.6) |
| Hispanic | 32 (40.5) | 13 (46.4) |
| African-American | 1 (1.3) | 0 (0.0) |
| BMI (SD), kg/m2 | | |
| Mean | 35.3 (10.8) | 36.4 (13.6) |
| Median | 33.3 | 32.7 |
| Range | 17.7–72.4 | 24.6–76.2 |
| BMI Category, no. (%), kg/m2 | | |
| < 25 | 9 (11.4) | 2 (7.1) |
| 25 to < 30 | 23 (29.1) | 8 (28.6) |
| ≥ 30 | 47 (59.5) | 18 (64.3) |
| Co-morbidities, no. (%) | | |
| Hypertension | 46 (58.2) | 16 (57.1) |
| Diabetes | 28 (35.4) | 11 (39.3) |
| Dyslipidemia | 27 (34.2) | 10 (35.7) |
| ≥ 2 co-morbidities, no. (%) | 31 (39.2) | 12 (42.9) |
| Infection type, no. (%) | | |
| Cellulitis | 68 (86.1) | 22 (78.6) |
| Abscess | 11 (13.9) | 6 (21.4) |
| Positive infection site cultures, no. (%) | 26 (32.9) | 13 (46.4) |
| Specific pathogen from positive cultures | | |
| MRSA | 10 (12.7) | 5 (17.9) |
| MSSA | 11 (13.9) | 4 (14.3) |
| Other Gram-positive | 5 (6.3) | 1 (3.6) |
| Primary DRG, no. (%) | | |
| 603 | 42 (53.2) | 28 (100) |
| 602 | 1 (1.2) | 0 |
| Other | 36 (45.6) | 0 |

accordingly: oritavancin failure followed by oral therapy with doxycycline plus ciprofloxacin as an outpatient, and post-discharge referral to a dermatology service for a non-infectious lesion.

Prior antibiotics leading to clinical failure are provided for both groups (Table 4). Almost 70% of antibiotic failures in cohort A prior to oritavancin treatment involved four oral agents commonly used for treatment of ABSSSIs (n, % of failures): trimethoprim/sulfamethoxazole (n = 19, 24.1%), clindamycin (n = 15, 19.0%), doxycycline (n = 11, 13.9%), and cephalexin (n = 9, 11.4%). In cohort B, patients failed the same four oral agents at a similar rate (n = 17, cumulative 60.7%; cohort A, n = 54, cumulative 68.4%).

**Table 2. Primary and secondary outcomes.**

| Parameter | Oritavancin (n = 79) | Comparator (n = 28) | Significance |
|---|---|---|---|
| Log-Transformed Length of Stay (days), 2016–2018 (mean, SD) a | 2.12 (0.086) | 2.59 (0.078) | p = 0.097 |
| Log-Transformed Length of Stay (days) by year (mean, no. of patients) a | 3.52, 9 | 2.00, 2 | p = 0.146 |
| 2016 | 1.89, 29 | 2.92, 9 | p = 0.037 |
| 2017 | 2.06, 41 | 2.51, 17 | p = 0.177 |
| 2018 | | | |
| Readmission Rate, no. (%) | | | |
| Within 30 days | 8 (10.1) | 17 (60.7) | p < 0.001 |
| Within 90 days | 10 (12.7) | 17 (60.7) | p < 0.001 |
| Average Cost per Hospitalization | | | |
| Actual cost, $ (SD) | 3,959.05 (2,377.34) | 4,256.28 (1,752.34) | p = 0.55 |
| Log-transformed cost, $ | 3,376.54 | 3,588.68 | p = 0.63 |
| Cost avoidance (index case plus readmission) b ($) | 207,423.64 | | |

a Oritavancin was adopted in DeTar formulary in August 2015 and data is omitted for this year.

b Cost avoidance per 100 patients treated with oritavancin is the sum of difference in hospital cost of index hospitalization based on log-transformed value of $212.11 plus difference in 90-day readmission rates using $3,879.43 average cost per hospitalization.

Abbreviation: SD, standard deviation.

There were 107 patient records available to review safety. There were no safety issues identified during the retrospective analysis in patients receiving oritavancin or alternative agents. There were no reports of infusion-related treatment-related adverse events or infusion discontinuations. There were no in-hospital deaths. Benefit derived from pre-medication with diphenhydramine 50 milligrams could not be evaluated since this practice was initiated from inception of the institutional protocol.

We conducted a simplified hospital cost avoidance analysis based on the difference in average cost per hospitalization (log-transformed) over the course of the 3-year time period between cohorts ($212.11, Table 2). For 100 patients, treatment with oritavancin based on our selective treatment strategy of early discharge and administration of oritavancin in an outpatient setting was associated with a cost avoidance of $21,211. The difference in observed

**Table 3. Characteristics, diagnosis and management of 6 patients without resolution of infection within 14 days post-dose oritavancin.**

| No. of patients | Description of infection at time of follow-up visit | Readmitted as Inpatient within 14 days (Yes/No) a |
|---|---|---|
| 1 | Unsatisfactory clinical response and was prescribed an oral combination of doxycycline and ciprofloxacin | No |
| 1 | Unsatisfactory clinical response to oritavancin; oral antibiotic therapy initiated in clinic also without resolution leading to amputation of 4th toe 14 days after oritavancin | Yes |
| 2 | Returned for a second dose of oritavancin, administered as outpatients, with resolution of lesion | No |
| 1 | Readmitted for additional parenteral antibiotic therapy for post-traumatic wound infection culture-positive for MSSA; patient responded to meropenem and additional gram-positive therapy, subsequently transferred to LTACH for completion of meropenem | Yes |
| 1 | Referred to dermatology for a non-infectious lesion | No |

a Infection-related readmissions are described for 2 patients within 14 days following administration of oritavancin. Not included in this table, an additional 6 patients requiring readmission to the hospital within 30 days (Table 2) post-dose oritavancin were managed for non-infectious reasons.

**Table 4. Previous antibiotic failures in order of occurrence in oritavancin group.**

| Antibiotic | Oritavancin (n = 79) a | Comparator (n = 28) b |
|---|:---:|:---:|
| Oral antibiotics | | |
| Trimethoprim/sulfamethoxazole | 19 | 6 |
| Clindamycin | 15 | 2 |
| Doxycycline | 11 | 4 |
| Cephalexin | 9 | 5 |
| Ciprofloxacin/levofloxacin | 6 | 1 |
| Minocycline | 3 | 1 |
| Cefdinir | 2 | 0 |
| Amoxicillin/clavulanate | 1 | 2 |
| Penicillin | 1 | 0 |
| Amoxicillin | 0 | 3 |
| Intravenous antibiotics | | |
| Ceftriaxone | 2 | 0 |
| Vancomycin | 2 | 0 |
| Meropenem | 1 | 0 |
| Ertapenem | 1 | 0 |
| Not recorded in medical chart/EHR | 21 | 7 |

a All 79 patients who received oritavancin failed prior outpatient treatment with oral or intravenous antibiotics; 73 antibiotic courses were recorded in 58 patients at discharge.

b Includes 28 patients who failed prior outpatient treatment with oral antibiotics; 24 antibiotic courses were recorded in 21 patients at discharge.

90-day readmission rates (oritavancin, 12.7%; comparator, 60.7%) multiplied by the average medical/surgical hospital stay per patient of $3,879.43 (DeTar Healthcare System) led to an additional cost avoidance of $186,212.64 per 100 patients. A total conservative cost avoidance with use of oritavancin was estimated as $207,423.64. The contribution of costs from diphenhydramine premedication to oritavancin therapy were not included as these were nominal.

## Discussion

Oritavancin is a long-acting lipoglycopeptide antibiotic that was studied in two Phase III randomized controlled trials. The Single-Dose Oritavancin in the Treatment of Acute Bacterial Skin Infections (SOLO I and SOLO II) trials demonstrated that a single 1200 mg intravenous (IV) dose of oritavancin was non-inferior to 7 to 10 days of IV vancomycin (1 g or 15 mg/kg twice daily) [9–11]. Furthermore, the pooled SOLO studies revealed similar patient outcomes of single-dose oritavancin (n = 392) compared to multidose vancomycin (n = 400) used to treat ABSSSIs entirely in the outpatient setting [12]. Efficacy response rates using a primary composite endpoint of early clinical evaluation were 80.4% and 77.5% for oritavancin and vancomycin, respectively.

The use of healthcare resources and costs associated with ABSSSI are largely driven by decision-making in the emergency department. A key study by Talan et al. highlights opportunities for transition of care to the outpatient arena in selected cases. In a prospective study of 12 US emergency departments in 2008 enrolling 619 adult patients with ABSSSI, 15.2% were admitted to an inpatient medical service. Common reasons for admission were need for intravenous antibiotics in 85.1% (and the only reason in 41.5%), surgery in 24.5%, and underlying disease in 11.7% [13]. Several studies have suggested that the Charlson Comorbidity Index (CCI), Eron Severity Score, and other well-defined criteria, based on the presence of comorbid

conditions and infection severity, can identify patients who may be candidates for effective and safe treatment as outpatients [14–19]. Use of such pathways for managing resource utilization in ABSSSIs can lead to cost savings through avoidable hospitalization.

Treatment failure is common with ABSSSIs, which subsequently adds significant healthcare costs. Lee et al conducted a prospective, observational study among 14 primary care clinics within the South Texas Ambulatory Research Network between 2007 and 2014; the primary outcome was treatment failure within 90 days of the initial visit. Overall, 21% (22/106) patients with *S. aureus* ABSSSIs experienced treatment failure [20]. Treatment failure with disease recurrence is multifactorial including non-adherence to prescribed antibiotics. Eells et al found that patient adherence with oral antibiotic therapy for an ABSSSI after hospital discharge was low (57%) and associated with poor clinical outcome in 46% of patients [21]. The combination of appropriate outpatient prescribing and the long acting effect of oritavancin may also contribute to decreased rates of antimicrobial resistance (AMR) by reducing antimicrobial pressure seen during hospitalization. As noted in a survey of young physicians by Gennero et al, it is widely recognized that AMR is a growing global issue but healthcare systems are lacking in addressing it at the local level [22]. In developing our local protocol, part of the aim was providing adequate education to providers on appropriate prescribing of oritavancin to promote antimicrobial stewardship (AS). Transitioning care to the outpatient setting reduces AMR well documented to occur with extended and frequent hospitalizations; while the long acting molecule affords assurance of completion of therapy, therefore, drastically reducing compliance as a cause for treatment failure.

In this study, the option of prescribing oritavancin in the appropriate patient with cellulitis or abscess led to shorter length of stay and lower hospital cost. While aLOS was non-significantly lower for years 2017 and 2018, the first 2016 year included only 11 patients. Over the study period, the aLOS was almost half a day shorter in cohort A. The lower 90-day readmission rate observed with oritavancin is even more remarkable compared with non-oritavancin antibiotics (12.7% versus 60.7%, respectively). The use of 90-day readmission served as a conservative estimate, although a 30-day readmission rate may be more reflective of antibiotic failure. However, cost avoidance was heavily weighted towards the excess costs of readmission observed in patients who received non-oritavancin antibiotics. We cannot account for physician decisions made at the point of care regarding hospital admission of patients in either treatment cohort.

Opportunities to decrease cost are invaluable to hospital administrators who observe increasing rates of ABSSSI and lack of consistent clinical management. The average length of stay prior to the advent of oritavancin utilization coupled with the allotted payment by most third-party payers decreases operating margins for hospitals. As reported by Ektare et al, hospital bed cost and average length of stay were the predominant factors in driving the total direct hospital costs for treating ABSSSI patients. In their analysis, the ability to move treatment of these patients to the outpatient setting demonstrated a potential cost savings of $13,090 to $13,473 by avoiding inpatient admission [23]. The results from this study represent a small community hospital in which Medicare is a large payer source, critical for small community hospitals. The results from this study demonstrate a benefit associated with the addition of oritavancin for the management of ABSSSI. Our findings lend further support to other non-academic community hospitals which do not serve as referral or tertiary care centers. As skin infections represent 2% of all US hospital admissions, increased use of oritavancin may represent an opportunity to recover several hundred thousand bed days for use with other patients [24].

This study has important limitations, including a retrospective methodology reliant on accurate medical record documentation and abstraction. Although categorization of diagnoses of skin infection by clinical type were made, the accuracy of these designations is unclear.

Severity of illness measurements and purulence of cellulitis cases were not available which could have served to differentiate between uncomplicated and complicated ABSSSIs. While assessment of disease severity may be more variable in real-world studies than in controlled ABSSSI clinical trials, abscess and cellulitis size restricted to $\geq$75 cm$^2$ is expected to be less stringently applied in this study [14]. Specific causes of antibiotic failure, such as medication adherence to oral and multi-dose therapies, could not be ascertained and antibiotic agents were not evaluated for either appropriate spectrum of in vitro activity according to confirmed microbiological cultures or appropriate dose. Therefore, we emphasize caution when interpreting these findings. Cohorts did not differ with respect to incidence of diabetes, and the use of one liter of 5% Dextrose in water required for admixture and infusion of oritavancin did not influence the selection of therapy in diabetics in this study. The failure rate and subsequent need for hospitalization in these patients reflect the ongoing observations at DeTar Healthcare System after 2018. However, future subset analyses could be valuable. The study protocol relied on medical chart abstraction to identify adverse events noted by caregivers but cannot preclude that treatment-emergent adverse events due to either oritavancin or other therapies were omitted, missed, or were deemed minor and not recorded. While diphenhydramine premedication is included in the protocol, there is no data to suggest if this practice should be adopted widely. Finally, there may be limited generalizability to other regions outside of South Texas. To our knowledge, this is the first study performed in a small community hospital comparing the impact of oritavancin against other historically accepted treatment options in patients who failed initial treatment.

More than 85% of US hospitals in 2019 are community-based, the majority of which have less than 200 beds and an average occupancy rate of 45 to 60% [25]. Our broad patient demographics present advantages to management of ABSSSIs which are not encountered in phase 3 clinical studies which often prevent a large and substantial number of patients eligible for enrollment. Real-world evidence (RWE) studies provide a bridge for validation or dispute between randomized controlled trials (RCTs) and clinical practice and filling current gaps in clinical knowledge. While RCTs provide evidence of efficacy, real-world studies produce evidence of comparative effectiveness, safety and economic performance in a naturalistic setting. Real-world studies are increasingly recognized by regulatory bodies such as the US Food and Drug Administration (FDA) [26]. Our data reflects the economic, efficacy and logistical impact of an oritavancin pathway seen by progressive reductions in average length of stay, readmission rates, and hospital costs. The results demonstrate the increased adoption of our institutional pathway in a small community hospital by healthcare providers over the 3 years of our study with realization of cumulative benefits over time.

## Acknowledgments

All coauthors were responsible for data interpretation and writing of this report. All authors had full access to all of the data in this study and take complete responsibility for the integrity of the data and accuracy of the data analysis.

**Authorship**

All named authors meet the International Committee of Medical Journal Editors (ICMJE) criteria for authorship for this article, take responsibility for the integrity of the work, and have given their approval for this version to be published.

**Additional contributions**

We would like to thank Mark Redell PharmD for his contribution to this publication. The expertise he provided in regards to the oritavancin molecule was a valuable addition to our study.

## Author Contributions

**Conceptualization:** Kimberly Saddler, Jason Zhang, Jennifer Sul, Miriams Castro-Lainez, Sheryl Kosler, Miguel Sierra-Hoffman.

**Data curation:** Kimberly Saddler, Jason Zhang, Jennifer Sul, Pruthvi Patel, Miriams Castro-Lainez, Mark L. Stevens, Sheryl Kosler, Emily Lowery, Miguel Sierra-Hoffman.

**Formal analysis:** Kimberly Saddler, Jason Zhang, Jennifer Sul, Miriams Castro-Lainez, Miguel Sierra-Hoffman.

**Investigation:** Kimberly Saddler, Jason Zhang, Miriams Castro-Lainez, Mark L. Stevens, Emily Lowery, Miguel Sierra-Hoffman.

**Methodology:** Kimberly Saddler, Jason Zhang, Jennifer Sul, Miriams Castro-Lainez, Mark L. Stevens, Emily Lowery, Miguel Sierra-Hoffman.

**Project administration:** Kimberly Saddler, Miguel Sierra-Hoffman.

**Resources:** Kimberly Saddler, Jennifer Sul, Pruthvi Patel, Miriams Castro-Lainez, Mark L. Stevens, Sheryl Kosler, Emily Lowery, Miguel Sierra-Hoffman.

**Software:** Kimberly Saddler, Jason Zhang, Miguel Sierra-Hoffman.

**Supervision:** Kimberly Saddler, Miriams Castro-Lainez, Miguel Sierra-Hoffman.

**Validation:** Kimberly Saddler, Jason Zhang, Jennifer Sul, Miriams Castro-Lainez, Miguel Sierra-Hoffman.

**Visualization:** Kimberly Saddler, Miriams Castro-Lainez, Miguel Sierra-Hoffman.

**Writing – original draft:** Kimberly Saddler, Jason Zhang, Jennifer Sul, Pruthvi Patel, Miriams Castro-Lainez, Mark L. Stevens, Sheryl Kosler, Miguel Sierra-Hoffman.

**Writing – review & editing:** Kimberly Saddler, Jason Zhang, Miriams Castro-Lainez, Miguel Sierra-Hoffman.

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
