## [Decision Letter · Decision Letter 0]

7 Jan 2021

PONE-D-20-39053

Improved Economic and Clinical Outcomes with Oritavancin Versus A Comparator Group for Treatment of Acute Bacterial Skin and Skin Structure Infections in a Community Hospital

PLOS ONE

Dear Dr. Kimberly Saddler,

Thank you for submitting your manuscript to PLOS ONE. After careful consideration, we feel that it has merit but does not fully meet PLOS ONE’s publication criteria as it currently stands. Therefore, we invite you to submit a revised version of the manuscript that addresses the points raised during the review process.

We look forward to receiving your revised manuscript.

Kind regards,

Francesco Di Gennaro

Academic Editor

PLOS ONE

Journal Requirements:

2. In the ethics statement in the manuscript and in the online submission form, please provide additional information about the patient records used in your retrospective study, including: a) whether all data were fully anonymized before you accessed them; b) the date range (month and year) during which patients' medical records were accessed; c) the date range (month and year) during which patients whose medical records were selected for this study sought treatment. If the ethics committee waived the need for informed consent, or patients provided informed written consent to have data from their medical records used in research, please include this information.

4.We note that you have indicated that data from this study are available upon request. PLOS only allows data to be available upon request if there are legal or ethical restrictions on sharing data publicly. For information on unacceptable data access restrictions, please see http://journals.plos.org/plosone/s/data-availability#loc-unacceptable-data-access-restrictions.

5.Thank you for stating the following in the Acknowledgments Section of your manuscript:

"The fees associated with publication of this study, including Rapid Service Fee

and Open Access fee, has been provided by Dr. Miguel Sierra-Hoffman, DeTar

Healthcare System, Victoria Emergency Associates, and the DeTar Family Medicine

Residency Program."

 "The authors received no specific funding for this work."

We note that one or more of the authors have an affiliation to the commercial funders of this research study : DeTar Healthcare System

Additional Editor Comments (if provided):

Dear Authors follow reviewer suggetions to improve your article

Reviewers' comments:

Reviewer's Responses to Questions

**Comments to the Author**

1. Is the manuscript technically sound, and do the data support the conclusions?

Reviewer #1: Yes

Reviewer #2: Partly

2. Has the statistical analysis been performed appropriately and rigorously? 

Reviewer #1: Yes

Reviewer #2: Yes

3. Have the authors made all data underlying the findings in their manuscript fully available?

Reviewer #1: Yes

Reviewer #2: Yes

4. Is the manuscript presented in an intelligible fashion and written in standard English?

Reviewer #1: Yes

Reviewer #2: Yes

5. Review Comments to the Author

Reviewer #1: In this work Authors analyzed the economic and clinical impact of a long-acting anti-gram positive antibacterial for the treatment of ABSSSIs.

To do this, they compared 79 patients received oritavancin who had failed previous outpatient antibiotic therapy (OPAT) with 28 patients receiving other antibiotics following OPAT failure and subsequent hospitalization.

This work is well performed and interesting to read. Attached there are several comments to improve the manuscript.

Introduction:

Lines 101 – 110: this is an interesting point. Authors should briefly describe the reason of this increase in healthcare costs of SSTIs despite the decrease of MRSA infections

Methods:

Authors use alternatively the acronyms ABSSSI and SSTI, however they refer to two different classification. I suggest defining better and clearly type of infections included in this study.

Results:

In this study the costs associated to antibiotics were not evaluated. However, if possible, I suggest also to consider costs associated with comparator antibiotics.

Discussion:

Use of long acting antibiotics that lead to early discharge of patients allows to reduce antimicrobial pressure in hospital setting. Reducing causes of antibiotic resistance is another important benefit of long-acting antibiotics. In my opinion this point and the need of including competencies in antibiotic use in all medical specialties should be briefly discussed (accordingly, this recent survey could be cited on this topic: “Italian young doctors' knowledge, attitudes and practices on antibiotic use and resistance: A national cross-sectional survey; J Glob Antimicrob Resist . 2020 Dec;23:167-173. doi: 10.1016/j.jgar.2020.08.022.”)

Reviewer #2: This article has some research value, but the scope of the topic is too large, or the scope of the research object is too small, it needs to be supplemented.The article only has the statistics data of hospitalized patients who had failed treatment.

6. PLOS authors have the option to publish the peer review history of their article (what does this mean?). If published, this will include your full peer review and any attached files.

Reviewer #1: No

Reviewer #2: No

---

## [Author Response · Author response to Decision Letter 0]

9 Feb 2021

Thank you for your review of our manuscript. Please see attached document "Response to Reviewers" for complete response.

---

## [Decision Letter · Decision Letter 1]

22 Feb 2021

Improved Economic and Clinical Outcomes with Oritavancin Versus A Comparator Group for Treatment of Acute Bacterial Skin and Skin Structure Infections in a Community Hospital

PONE-D-20-39053R1

Dear Dr. Saddler,

We’re pleased to inform you that your manuscript has been judged scientifically suitable for publication and will be formally accepted for publication once it meets all outstanding technical requirements.

Kind regards,

Francesco Di Gennaro

Academic Editor

PLOS ONE

Additional Editor Comments (optional):

dear authors congratulations

Reviewers' comments:

Reviewer's Responses to Questions

**Comments to the Author**

1. If the authors have adequately addressed your comments raised in a previous round of review and you feel that this manuscript is now acceptable for publication, you may indicate that here to bypass the “Comments to the Author” section, enter your conflict of interest statement in the “Confidential to Editor” section, and submit your "Accept" recommendation.

Reviewer #1: All comments have been addressed

Reviewer #2: All comments have been addressed

2. Is the manuscript technically sound, and do the data support the conclusions?

Reviewer #1: Yes

Reviewer #2: Yes

3. Has the statistical analysis been performed appropriately and rigorously? 

Reviewer #1: Yes

Reviewer #2: Yes

4. Have the authors made all data underlying the findings in their manuscript fully available?

Reviewer #1: Yes

Reviewer #2: Yes

5. Is the manuscript presented in an intelligible fashion and written in standard English?

Reviewer #1: Yes

Reviewer #2: Yes

6. Review Comments to the Author

Reviewer #1: (No Response)

Reviewer #2: Thank you for your replies and hard work on the manuscript. I accept your interpretations, and also hope your future publications will have more considerations.

7. PLOS authors have the option to publish the peer review history of their article (what does this mean?). If published, this will include your full peer review and any attached files.

Reviewer #1: No

Reviewer #2: No

---

## [Editor Report · Acceptance letter]

25 Feb 2021

PONE-D-20-39053R1 

Improved Economic and Clinical Outcomes with Oritavancin Versus A Comparator Group for Treatment of Acute Bacterial Skin and Skin Structure Infections in a Community Hospital 

Dear Dr. Saddler:

I'm pleased to inform you that your manuscript has been deemed suitable for publication in PLOS ONE. Congratulations! Your manuscript is now with our production department. 

Kind regards, 

on behalf of

Dr. Francesco Di Gennaro 

Academic Editor

PLOS ONE